# Chemoprevention for the Populations of Malaria Endemic Africa

**DOI:** 10.3390/diseases10040101

**Published:** 2022-11-08

**Authors:** Brian Greenwood, David Schellenberg

**Affiliations:** Faculty of Infectious and Tropical Diseases, London School of Hygiene and Tropical Medicine, Keppel St., London WC1E 7HT, UK

**Keywords:** malaria, chemoprevention, sub-Saharan Africa

## Abstract

Drugs have been used to prevent malaria for centuries, but only recently have they been used on a large scale to prevent malaria in the resident population of malaria endemic areas in sub-Saharan Africa. This paper discusses some of the reasons for the hesitancy in adoption of chemopreventive strategies in sub-Saharan Africa, reasons why this hesitancy has been overcome in recent years and the range of target groups now identified by the World Health Organization as those who can benefit most from chemoprevention. Adoption of carefully targeted chemopreventive strategies could help reverse the recent stagnation in the decline in malaria in sub-Saharan Africa that had been taking place during the previous two decades.

## 1. Introduction

Drugs have been used to prevent malaria in the resident population of malaria endemic areas for centuries. However, only relatively recently has the World Health Organization recommended the large-scale use of antimalarials to prevent malaria in residents of these countries. This commentary discusses some of the reasons why there was a reluctance to adopt this approach to malaria control forty years ago and how a change in this attitude to the deployment of chemopreventive malaria strategies in the resident population of endemic countries by international and local health authorities has come about.

## 2. Early Studies

Malaria prophylaxis with cinchona bark (Jesuit’s powder) was initially restricted to a small population outside its original home in Peru because its source was carefully guarded and it was expensive [1]。 The development of methods for extracting quinine, and establishment of cinchona plantations outside South America, made quinine more accessible and less expensive. However, its use in endemic populations as a prophylactic was largely restricted to special groups such as the workers who built the Panama Canal and the populations highly exposed to malaria during the Italian malaria eradication campaigns [2]. Development of synthetic antimalarials in the middle of the twentieth century encouraged a wider use of antimalarials for prophylaxis in endemic populations, as well as in expatriates including the military. The efficacy of anti-malarial medicines when used in the resident population of malaria endemic areas in sub-Saharan Africa was established in a number of well conducted clinical trials of chemoprophylaxis in the 1950s and 1960s. These trials demonstrated convincingly that malaria chemoprophylaxis reduced clinical attacks of malaria, anaemia and days lost from work or school [3,4,5]. At this time, some companies provided antimalarials to their work force, appreciating its economic benefit in preventing days lost from work, and chloroquine was administered widely to school children during the high malaria transmission season in Senegal to reduce days lost from school. Antimalarials such as pyrimethamine were openly marketed in many cities in sub-Saharan Africa, including Nigeria where pyrimethamine (Daraprim) was advertised widely as a ‘Sunday Sunday’ medicine and used by those who could afford it because of its recognised benefit. A few modest sized pilot implementation studies were undertaken which demonstrated the ability of community workers to administer chemoprophylaxis effectively, including the Danfa project in Ghana in the 1970s [6], a study in Haute-Volta (Burkina Faso) in the early 1980s [7] and a trial in The Gambia in the mid-1980s [8]. Mass drug administration (MDA) programmes involving the whole population which aimed at interrupting transmission, for example the Garki project in Nigeria in the early 1970s [9], showed marked reductions in cases of malaria but these programmes were not sustained and no malaria endemic country in sub-Saharan Africa adopted any form of chemoprevention as a national policy at that time. Interest in the potential benefits of chemoprevention in the resident population of malaria endemic areas in sub-Saharan Africa, both internationally and nationally, gradually frittered away. China was an exception with widespread deployment of several synthetic antimalarial compounds prophylactically, largely through MDA programmes, in the 1950s and 1960s [10].

## 3. Resistance to the Widespread Use of Chemoprevention in Malaria Endemic Populations

Why was chemoprevention for the resident population of highly malaria endemic countries not taken up widely despite its proven efficacy in travellers and expatriates resident in malaria endemic areas? Initial resistance to this approach came from several distinguished and well-informed malaria experts who recommended that antimalarials should not be used widely in the local population of malaria endemic countries for prevention and their use for prevention restricted to non-immune travellers and expatriates resident in these areas. Some of their reasons for adopting this position, which included concerns over generation of drug resistance, impairment of immunity, safety and cost, are summarised in Table 1.

These concerns were widely accepted by most international organisations supporting malaria control, including WHO, and during the 1970s and 1980s there was resistance from WHO to the use of large-scale malaria chemoprevention in the local population of malaria endemic countries in Africa.

## 4. Increasing Acceptance of the Potential Value of Chemoprevention in Malaria Endemic Populations

The one exception to the use of malaria chemoprevention among the population of malaria endemic populations in the 1970s and 1980s was its use in pregnancy, with chemoprophylaxis with chloroquine being encouraged. However, this was little used because of poor acceptance by pregnant women and the staff of antenatal clinics. A breakthrough came in 1994, with a report from Malawi that administration of two treatment doses of sulphadoxine/pyrimethamine (SP) during the second and third trimesters of pregnancy, respectively, was more effective in preventing placental malaria than chloroquine chemoprophylaxis [11], and a subsequent study in Kenya showed that administration of SP during pregnancy reduced maternal anaemia [12]. On the basis of this very limited evidence, in 1998, WHO’s Malaria Expert Committee recommended administration of treatment doses of SP to women in their first and second pregnancies on two occasions as part of national malaria control programmes in areas of sub-Saharan Africa where pregnant women were at medium or high risk of malaria infection [13]. Because women were provided with protection from malaria for only a limited period of their pregnancy, this approach acquired the name of Intermittent Preventive Treatment of Malaria in Pregnancy (IPTp)(Figure 1).

This cautious approach countered concerns about the feasibility of delivering chemoprophylaxis, the likelihood that chemoprophylaxis would lead to resistance, the costs of regular prophylaxis and the possibility that the development and maintenance of naturally acquired immunity to malaria would be impaired. However, increasing recognition of the illogicality of providing protection against malaria for just a part of pregnancy, leaving women at risk during much of their pregnancy, led WHO to expand progressively the scope of its recommendations on ITPp. In 2004, based on sound evidence that three doses of IPTp were more effective than two [14], WHO recommended three doses of IPTp in the first and second pregnancies. In 2012, the recommendation was changed to one of administration of monthly doses after the first trimester (potentially up to 6 doses), based on the timing of recommended contacts with antenatal care services, with the aim of achieving protection throughout the second and third trimesters [15].

The potential value of malaria chemoprevention was also evaluated in the mid-1990s among infants living in areas of high, perennial malaria transmission who were at high risk of severe malaria, including severe anaemia, and death. A key initial study in Tanzania demonstrated that sustained chemoprophylaxis in infancy reduced the incidence of clinical malaria and severe anaemia [16]. However, just as with IPTp, there were concerns about its potential impact on the development of resistance, its potential interference with the development of naturally acquired immunity, safety and costs of providing chemoprophylaxis routinely to all infants living in intense transmission settings. This apprehension came at a time when interest and investments in malaria control were at a low ebb, when few malaria medicines were available, the malaria drug development pipeline almost non-existent and before the creation of the Global Fund to fight AIDS, TB and malaria. Nevertheless, given the substantial benefits of chemoprophylaxis and the extremely high ongoing burden of malaria in young children, a study was designed to assess whether intermittent preventive treatment of malaria in infants (IPTi) could overcome the challenges of prophylaxis without losing all of its benefits by administering antimalarial drugs to children attending routine vaccinations during the first year of life. The initial study showed that administration of SP on three occasions at the time of vaccination during the first year of life led to a 59% reduction in clinical episodes of malaria and a 50% reduction in the incidence of severe anaemia [17]. These results sparked interest among researchers and to an informal meeting of UNICEF and WHO to review the results, identify key research gaps, and to consider the designs and duration of studies needed to address these gaps. Subsequent trials, co-ordinated through an IPTi consortium, showed the potential of this approach to reduce clinical malaria during the first year of life by about 30% [18], with each dose protecting children for 21–42 days [19]. This evidence led to the recommendation by WHO in 2010 for the introduction of IPTi in areas of medium or high malaria endemicity where SP was still effective [20]. However, despite this WHO recommendation, IPTi was not adopted widely, perhaps partly due to the perceived low efficacy of the approach and the perception of SP as a failed drug [21].

Children living in areas where malaria transmission is restricted to just a few months each year are less likely to benefit from IPTi than children resident in areas where there is a substantial risk of malaria throughout the year. Recognition of this fact led to the development of a further chemopreventive strategy, Intermittent Preventive Treatment of malaria in children (IPTc), now termed Seasonal Malaria Chemoprevention (SMC), in which SP and amodiaquine (AQ) are given to young children at monthly intervals during the high malaria transmission season. In 2012, based on the results of several clinical trials which showed up to an 85% reduction in clinical episodes of malaria [22], SMC became the third major chemopreventive strategy to be recommended by WHO [23]. The 2012 WHO recommendation limited SMC to children up to 5 years old living in the Sahel sub-region of Africa, and was interpreted by some to restrict the intervention to a maximum number of four rounds of treatment in any one season. Despite these limitations, and the need to establish a novel platform to deliver monthly doses of SMC during the rainy season, this strategy has been widely deployed and, in 2020, reached over 30 million children in 13 countries where malaria transmission is limited to a few months each year [24], preventing many cases of malaria and many deaths [25]. The rapid uptake of SMC has probably been due to the high efficacy and estimated cost effectiveness of the strategy, and the visible benefits for households, leading to high acceptability of SMC by local communities.

Two additional uses of malaria chemoprevention have also been investigated in recent years. A number of studies have highlighted the burden of malaria in school-age children, in whom malaria causes clinical illness, anaemia and impaired educational outcomes [26]. In addition, this age group makes an important contribution to sustaining transmission of the infection [27]. Several approaches to the delivery of chemoprevention in school-age children have been evaluated, confirming significant protection from malaria infection, clinical malaria and anaemia and identifying these children as a potential target for chemopreventive strategies [28].

A second new use case addresses the very high mortality among children admitted to hospital with severe anaemia or malnutrition during the first few months after their discharge from hospital. It has recently been demonstrated that in the case of children admitted to hospital for treatment of severe anaemia, malaria chemoprevention can significantly reduce death and hospital admissions during the three months after hospital discharge [29,30,31].

## 5. Updated WHO Malaria Chemoprevention Guidelines

In 2022, WHO updated its guidelines on the chemoprevention of malaria for people living in endemic settings [32]. These guidelines were the product of a formal guideline development process launched in 2020, which drew on systematic reviews of the evidence on the efficacy and safety of IPTp, IPTi, SMC, MDA, chemoprevention in school-age children (IPTsc) and chemoprevention after discharge from hospital following an admission with severe anaemia (Post-Discharge Chemoprevention—PDMC). The decision by WHO to review old guidelines and to develop new guidelines was prompted by a combination of factors. Firstly, new data were available for all chemopreventive strategies. Secondly, some of WHO’s previous recommendations were considered overly restrictive. For example, the IPTi recommendation restricted SP administration to three doses in the first year of life, two of them before the age of six months. This did not permit the targeting of age groups most frequently affected by severe disease in many settings. In addition, growing experience with IPTp and SMC enabled cross-comparisons between chemopreventive strategies, for example in relation to the effect of drug resistance on the efficacy of chemoprevention, and the effect of chemoprevention on resistance [33]. Finally, the growing appreciation that progress in malaria control had stalled prompted the need to review how best to use existing tools. The malaria community had long recognised the need to enhance its problem-solving approaches, rather than applying the same control tools everywhere, but the restrictive way in which the chemoprevention guidelines were initially formulated made it difficult for countries to adapt recommended strategies to suit their individual settings.

The updated WHO guidelines are available online [32] and provide not only the recommendations but summaries of the evidence upon which they are based, the evidence-to-decision process, practical and other considerations, including outstanding research needs. The following key changes were made to existing guidelines —

a.The IPTp recommendation was expanded to cover administration of SP to pregnant women of all gravidities, at predetermined intervals, to provide sustained protection throughout the second and third trimesters of all pregnancies.b.The IPTi recommendation was made more permissive, opening the door for additional doses and implementation beyond 12 months of age. Hence, the strategy was renamed Perennial Malaria Chemoprevention (PMC).c.The SMC strategy was also made more permissive, removing restrictions on the age group to whom SMC could be given and the number of rounds of treatment that could be given each year.

In addition to these amendments to existing recommendations, three additional sets of recommendations on the potential use of chemopreventive strategies in the population of malaria endemic countries were made. These related to —

a.Intermittent preventive treatment in school-age children (IPTsc). School-aged children living in malaria-endemic settings with moderate to high perennial or seasonal transmission can be given a full therapeutic course of antimalarial medicine at predetermined times as chemoprevention, to reduce disease burden. However, the recommendation included the proviso that this should only be considered if resources allow for its introduction without compromising chemoprevention interventions for those age groups carrying the highest burden of severe disease, such as children under five years of age.b.Post Discharge Malaria Chemoprevention (PDMC). Children admitted to hospital with severe anaemia living in settings with moderate to high malaria transmission should be given a full therapeutic course of an antimalarial medicine at predetermined times following discharge from hospital to reduce re-admission and death.c.Mass drug administration (MDA). A set of recommendations was made to support the use of MDA in moderate and high transmission settings to reduce disease burden, including in specific situations such as emergencies or epidemics of febrile illnesses, and to reduce *Plasmodium falciparum* and *P. vivax* transmission in very low and low transmission settings. Although a community parasite prevalence of 10% is recognised as the boundary between low and moderate transmission [32], all recommendations on chemoprevention consider this an indicative threshold which should not be regarded as an absolute criterion for determining the applicability of a strategy. In the case of MDA, it is biologically plausible that in settings near the 10% threshold, MDA may reduce both disease burden and transmission.

## 6. The Reasons Underlying the Change in Attitude of WHO to Chemoprevention in the Population of Malaria Endemic Countries

As indicated above, the ways in which chemopreventive strategies could be used to reduce the burden of malaria in the population of endemic countries has recently been expanded and it is, therefore, important to consider why the concerns over the use of chemopreventive strategies in the resident population of malaria endemic countries expressed in the 1970s and 1980s have not been realised. The developments that have led to the renewed interest in the potential of chemopreventive strategies in malaria endemic countries and which have overcome of these initial concerns are summarised in Table 2.

## 7. Conclusions

Increasing recognition of the potentially valuable role that chemoprevention can play in protecting segments of the population of malaria endemic countries from malaria highlights the need to strengthen operational capacities for delivering chemopreventive strategies and to continue research in this field.

Strengthening capacity to generate and use local data will ensure that the age groups experiencing most of the severe disease, and the extent of seasonal variation in transmission within a country, or region within a country, are known and can determine where and when a particular chemopreventive strategy might be appropriate. National health information systems need to capture relevant information with sufficient detail, and malaria control programmes need access and capacity to use these data to inform their programmatic decisions and to monitor changes in malaria epidemiology over time. Evaluation of on-going programmes on a regular basis is needed to evaluate efficacy and effectiveness and to detect early emergence of resistance to the drug(s) being employed in a chemopreventive programme.

Key research needs specific to individual recommendations are captured (under ‘More Info’) in the WHO guidelines [32]. In addition, research is needed to determine the situations in which ‘rebound malaria’ on stopping chemoprevention is likely to be a significant risk and what could be done to mitigate this risk. Research and development efforts need to specifically consider the development of malaria drugs with a long half-life that makes them suitable for chemoprevention, as well as for treatment.

Introduction of an appropriate, targeted chemopreventive strategy into a national malaria control programme has been shown to be feasible and effective. However, as recognised by its initial sceptics, chemopreventive programmes are demanding to deliver, require frequent contacts between the recipient and the drug deliverer, and achieving a high level of coverage can be challenging, as seen in the case of IPTp. In addition, new drugs and drug combinations are needed that could replace the old drugs currently being used for chemoprevention when and where these become needed. For these reasons, provision of a high level of protection to the at-risk population of malaria endemic countries through an immunological rather than a pharmacological approach becomes a potentially attractive option as this may be easier to deliver and more sustainable. The 2021 WHO recommendation for widescale use of the first malaria vaccine, RTS,S/AS01 [34], opens the door to a new paradigm in malaria control. However, its limited efficacy and availability emphasise the need to continue to explore other avenues. Additional vaccines that can provide at least the same level of protection as chemopreventive strategies, and ideally provide protection over a period of many years, are needed. In places where the burden of malaria remains stubbornly high, it may be necessary to combine chemoprevention and vaccination on top of existing strategies to achieve a high level of malaria control [35].

The cautious approach to malaria chemoprevention in endemic country populations has led to a focus on situations where protection is required for only a limited period, for example during pregnancy or seasonal malaria transmission. Monoclonal antibodies [36], which can provide up to a six-month period of protection following a single injection, provide a potential immunological approach to the prevention of malaria in situations where chemopreventive strategies are currently being promoted. A time may come when chemopreventive strategies are no longer needed and can be replaced by active or passive immunological approaches. However, until this time is reached, chemopreventive strategies appropriately deployed, have the potential to play an important role in reversing the stagnation in progress towards malaria control currently being seen in many high burden countries.

## Figures and Tables

**Figure 1 diseases-10-00101-f001:**
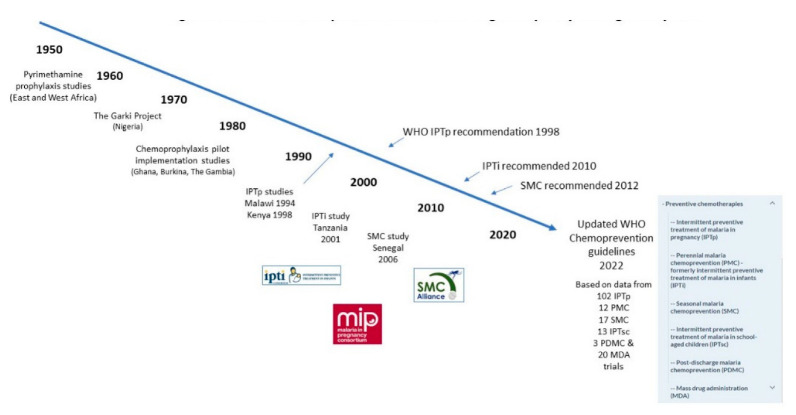
Malaria chemoprevention research & global policy through the years.

**Table 1 diseases-10-00101-t001:** Historic concerns expressed over the use of chemoprevention in the resident population of malaria endemic areas.

The limited number of anti-malaria drugs:	The limited number of antimalarial drugs available for treatment and limited investment in new drug development raised concerns over using these drugs for prevention as opposed to treatment.
Drug resistance:	Widespread use of antimalarial drugs in the resident population would accelerate the development of resistance to the relatively small number of available antimalarials that could be used for treatment or prophylaxis in non-immune travellers and expatriate residents.
Impairment of immunity:	Prophylactic use of antimalarials in young children would impair the development of naturally acquired immunity by preventing frequent malaria infections and thus put them at high risk of malaria when they stopped taking the antimalarials.
Safety, tolerability:	Tolerability and acceptability issues would emerge if drugs needed to be taken regularly over a long period, and safety could become an issue even for drugs which only rarely caused serious side effects if given to large numbers of people who at the time they received the drug were not infected or who were asymptomatic.
Administration:	It would not be possible to deliver prophylactic drugs at scale, especially to young children, as there was no established delivery system.
Cost:	Large scale administration of antimalarials prophylactically was not affordable; the limited funds available for malaria control would be used for treatment, including presumptive treatment of malaria in situations where parasitological diagnosis of malaria was not available.

**Table 2 diseases-10-00101-t002:** Validity of the initial concerns over the use of chemoprevention in the resident population of malaria endemic areas.

Drug resistance:	Widespread use of IPTp with SP may have contributed to the expansion of SP resistance in *Plasmodium falciparum* in southern and eastern Africa but, after many years of use, IPTp with SP still provides useful protection in many settings.
Impairment of immunity:	A limited number of studies have shown ‘rebound’ after a chemoprevention programme has been halted, but this has been uncommon and all studies have shown that the protection achieved during the period of chemoprevention outweighed any enhanced risk in the subsequent period.
Safety, tolerability and acceptability:	There have been no major safety issues with IPTp, IPTi, SMC or MDA programmes with risk/benefit analyses strongly favouring the interventions. Tolerability has not proved a major issue, even with amodiaquine, because of the perceived benefit of the intervention by the population.
Administration:	This was perceived as a particular problem when delivery was required outside the established delivery system, such as antenatal or vaccination clinics. However, experience with SMC has shown high coverage levels can be achieved using paid or volunteer community health workers and additional contacts at EPI and ANC clinics has improved potential coverage based on these platforms.
Cost:	This concern was probably justified at the time concerns were expressed but the increase in both national and international financial support for malaria control has reduced the validity of this concern. There is now a wider range of inexpensive antimalarials with proven effectiveness and which are easy to deliver than at any time in the past. Cost effectiveness estimates of chemopreventive strategies are very favourable when compared to other malaria control tools considered highly or very highly cost effective.

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
