# Peer review of "Chemoprevention for the Populations of Malaria Endemic Africa"

_diseases, 2022, doi:10.3390/diseases10040101_

Round 1

Reviewer 1 Report

Include tables 

Author Response

Tables included.

Reviewer 2 Report

The commentary article entitle" Chemoprevention for the populations of malaria endemic African" is very well written by the authors. The introduction and information regarding reluctance to adopt chemoprevention approch to malaria control 40 years ago and how a change in this attitude to the deployment of chemopreventive strategies now is excellently discussed by the authors. This commentary is highly informative and highly recommend ed read. I strongly, recommend this commentary article to publish in Diseases without any further changes.

Author Response

We thank the reviewer for his appreciative comments.

Reviewer 3 Report

Excellent commentary review. The authors can further improve their manuscript by adding a graphical or tabular presentation of the chronological order of important  events.

Author Response

A figure has been added as suggested and some minor typos corrected.